# Late Ordovician High Ba-Sr Intrusion in the Eastern North Qilian Orogen: Implications for Crust–Mantle Interaction and Proto-Tethys Ocean Evolution

**Shaoqing Zhao** [1,*], **Lianfu Hai** [2,3,*], **Bin Liu** [1], **Huan Dong** [1], **Chao Mei** [2,3], **Qinghai Xu** [1], **Caixia Mu** [2,3] and **Xiangcheng Wei** [2,3]

1    School of Geosciences, Yangtze University, Wuhan 430100, China; binliu@yangtzeu.edu.cn (B.L.)
2    Mineral Geological Survey Institute of Ningxia Hui Autonomous Region, Yinchuan 750021, China
3    Institute of Mineral Geology of Ningxia Hui Autonomous Region, Yinchuan 750021, China
*    Correspondence: shaoqing@yangtzeu.edu.cn (S.Z.); hailianfu@163.com (L.H.)

**Abstract:** High Ba-Sr granitic rocks are widespread in Phanerozoic orogenic systems, and their petrogenesis is important for revealing the evolutionary process of the Proto-Tethys Ocean in the North Qilian orogenic belt. This paper presents a combination of zircon U-Pb age, whole-rock major and trace element concentrations, and Sr-Nd-Hf isotopic data for Caowa high Ba-Sr dioritic intrusion from the eastern part of the North Qilian orogenic belt, aiming to decipher its petrogenesis and tectonic setting. LA-ICP-MS zircon U-Pb dating yield an emplacement age of $450 \pm 2$ Ma for the Caowa intrusion, indicating a magmatic activity of the Late Ordovician. The Caowa quartz diorites contain moderate contents of $SiO_2$, $MgO$, $Mg^{\#}$, and resultant high concentrations of $Na_2O +$ $K_2O$, $Fe_2O_3^{T}$, and $Al_2O_3$, displaying calc-alkaline and metaluminous characteristics. The studied samples have relatively elevated Ba (up to 1165 ppm) and Sr (561 to 646 ppm) contents, with obvious enrichment in LILEs (e.g., Ba, Th, U) and depletions in HFSEs (e.g., Nb, Ta, Ti), resembling those of typical high Ba-Sr granitoids in subduction zones. Together with enriched Sr-Nd isotopic composition $[(^{87}Sr/^{86}Sr)_i = 0.7082–0.7086, \varepsilon_{Nd}(t) = -5.1$ to $-4.9]$, and the wide ranges of zircon $\varepsilon_{Hf}(t)$ values $(-13.2$ to $+8.5)$, it suggests that these high Ba-Sr quartz diorites were derived from a mixture magma source between the ancient crust materials and the enriched lithospheric mantle metasomatized by fluid released from subducted oceanic crust or sediment. Taking into account the ophiolites, high pressure metamorphic rocks, and arc magmatic rocks in the region, we infer that due to the influence of the northward subduction of the Qilian Proto-Tethys Ocean, the Laohushan oceanic crust of the North Qilian back-arc basin was subducted during the Late Ordovician and resulted in extensive metasomatism of lithospheric mantle by fluids derived from oceanic crust or sediments, and the Caowa high Ba-Sr quartz diorites were generated in the process of crust–mantle interaction during the Late Ordovician.

**Keywords:** high Ba-Sr granitoids; Late Ordovician; subduction; Proto-Tethys; North Qilian orogen



## 1. Introduction

The Qilian orogenic system, located in the northeastern Tibetan Plateau, has been considered as the northernmost orogenic collage of the Proto-Tethys Realm. It records a multistage tectonic process from the Neo-proterozoic continental breakup of the Rodinia supercontinent, the Early Paleozoic oceanic subduction and accretion, finally resulting in arc-continent and continent collisions [1–3]. As a significant unit of the Qilian orogenic system, the North Qilian orogenic belt has been confirmed by the typical subduction–accretion suture with a complex trough–arc–basin system, marking the tectonic evolution of the Proto-Tethys Ocean [4–6]. Previous studies have been carried out on the outcrops of the south mid-ocean-ridge-type ophiolites, the middle part subduction/collision-related arc igneous rocks, and the north back-arc-basin-type ophiolites in the North Qilian, and indicate

that the northward subduction of the Qilian Proto-Tethys Ocean during the Ordovician is the most popular viewpoint [5,7,8]. Nevertheless, the evolutionary process of the Proto-Tethys Ocean in the eastern part of North Qilian belt remains unclear. Many Early Paleozoic adakitic granitoids are recognized in the Eastern North Qilian orogenic belt, and some recent studies suggested that they generated during continent–continent collision or post-collision collapse [9–13]. However, their petrogenesis and geodynamic evolution processes are still disputed, and little research has focused on the oceanic subduction-related granitoids (e.g., I-type granite, high Ba-Sr granitic rocks) in the region, especially in the Nanhuashan area of Ningxia province, easternmost North Qilian belt.

High Ba-Sr granitoids, as a distinct group of magmatic rocks, are widespread in Phanerozoic orogenic systems and provide a large amount of effective information to estimate the tectonic evolution process of the orogenic belts [14–17]. Compared to traditional I-, S-, and A-type granitoids, the high Ba-Sr granitoids have several unique signatures such as alkali-rich, high Ba (>500 ppm) and Sr (>300 ppm) contents, and low Rb (<200 ppm) and Y (<30 ppm) contents with Rb/Ba ratios < 0.2. They also display high Sr/Y ratios, enrichment of light rare earth elements (LREEs) and large-ion lithophile elements (LILEs), depletion in heavy rare earth elements (HREEs) and high field-strength elements (HFSEs), with no significant negative Eu anomalies [14,18]. High Ba-Sr granitoids always carry geochemical and isotopic signatures of enriched mantle sources, which is linked to the oceanic subduction-related metasomatism [15,19–23]. However, the possible mechanisms for the generation of high Ba-Sr granitoids, such as partial melting of subducted ocean islands/ocean plateaus [14], melting of mafic lower crust [18,24], or magma mixing [16,25], have also been proposed. Therefore, the recognition of high Ba-Sr granitoids may provide particular information on crust–mantle interactions and the growth of the continental crust in subduction zones.

In this paper, we present zircon U-Pb geochronology, whole-rock geochemistry, and Sr-Nd-Hf isotopic data for the Late Ordovician dioritic intrusion (mainly composed of quartz diorites) with high Ba-Sr signatures in the Nanhuashan area of Ningxia, Eastern North Qilian orogenic belt (Figure 1). The results, together with previously published data, are used to elucidate the petrogenesis of the high Ba-Sr quartz diorites, and further evaluate their geodynamic implications for the Proto-Tethys Ocean evolution.

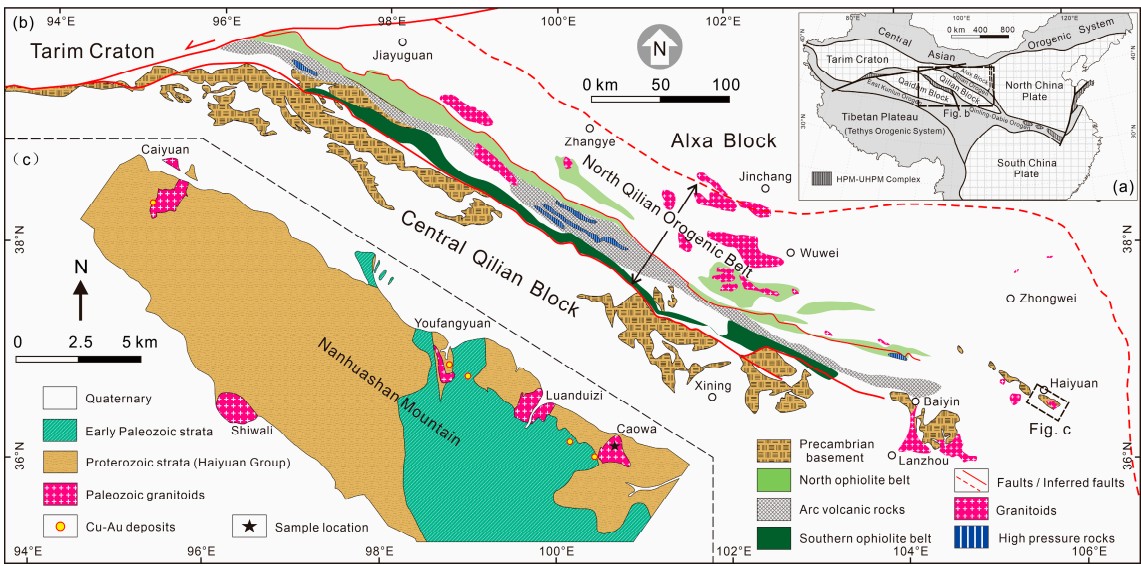

**Figure 1.** (**a**) Simplified geological map of China (modified from [5]); (**b**) simplified geological map of the North Qilian Orogenic Belt showing distributions of the main tectonic units (modified from [26]), and (**c**) simplified distribution map of the Caowa dioritic intrusion, showing the studied samples' location.

## 2. Geological Background and Petrography

The Qilian orogenic belt, located in the northeastern margin of the Tibetan Plateau, is divided into three units from north to south: the North Qilian orogenic belt (NQOB), the Central Qilian block, and the South Qilian accretionary complex belt [1,27] (Figure 1a). The NQOB, extending NW–SE more than 1000 km, lies between the Alxa Block to the north and the Central Qilian Block to the south, and is separated from the Altyn-Tagh Fault to the west (Figure 1b). It is dominated by an outcrop of Early Paleozoic ophiolites (MORB-type), high-pressure metamorphic rocks (e.g., eclogites, blueschists), and a series of subduction/collision-related magmatic rocks, and was confirmed as a typical subduction–accretion orogenic belt with a complex trough–arc–basin system [5,6]. Two ophiolite sequences are distributed in the NQOB (Figure 1b). The southern ophiolite belt (550−496 Ma) is connected with Aoyougou, Yushigou, and Dongcaohe ophiolites and mainly consists of mantle peridotite, cumulate gabbro, and MORB, which document the oceanic crust fragments of the Qilian Proto-Tethys Ocean [5]. The northern ophiolite belt (490−448 Ma) is a typical back-arc basin ophiolite (e.g., Jiugequan, Biandukou, and Laohushan ophiolites), representing the extension of the North Qilian back-arc oceanic basin [5,26,28]. There are volcanic-magma arc belt (520−440 Ma) outcrops between the two ophiolite belts, mainly including mafic and felsic volcanic rocks [8]. The high pressure–low temperature (HP–LT) metamorphic rocks in the NQOB are predominantly composed of blueschists and low-temperature eclogites, with metamorphic ages ranging from 490 to 440 Ma [2,29,30]. In addition, a large number of Early Paleozoic granitoids (520−420 Ma), including adakitic, I-, and A-type granitoids, have been recognized in the NQOB from the Changma-Dachadaban-Corridor Nanshan in the west to the Leigongshan-Laohushan-Quwushan Mountain in the east, and their formation was linked to oceanic subduction, closure, and post-collision processes of the Qilian Proto-Tethys Ocean [10,12,13,31,32].

The Nanhuashan Mountain, connecting with Quwushan-Baojishan-Laohushan-Leigongshan Mountain to the west in the eastern part of NQOB, exposed a large amount of Early Paleozoic granitic intrusions. These intrusions, including Caiyuan, Shiwali, Youfangyuan, Luanduizi, and Caowa intrusions, intrude into the Meso-proterozoic Haiyuan Group. Notably, these magmatic events were very closely related to Cu-Au mineralization in the Nanhuashan area. The high Ba-Sr signature rocks reported in this paper were collected from the Caowa intrusion (Figure 1c). The Caowa dioritic intrusion is mainly composed of quartz diorites, with an outcrop area of ~4 km$^2$, which are medium- to coarse-grained rocks (Figure 2a). The quartz diorites are mainly composed of plagioclase (40–50 vol.%), hornblende (20–30 vol.%), quartz (5–10 vol.%), and K-feldspar (5–10 vol.%), with minor amounts of iron oxides, zircon, and apatite (Figure 2b–d). The plagioclase generally forms euhedral–subhedral laths with polysynthetic twinning. Some of these laths display concentric compositional zoning and are more sericitized (Figure 2b,d). Hornblende occurs as euhedral–subhedral grains, with length of 0.5–1 mm, and usually appears as a mafic polycrystalline agglomerate (Figure 2c,d). K-feldspar is subhedral–anhedral and mainly consists of orthoclase and occasional microcline. Quartz is anhedral and fills the interstices between amphibole and plagioclase crystals. In addition, acicular apatite commonly occurs near the hornblende polycrystalline agglomerate (Figure 2c,d).

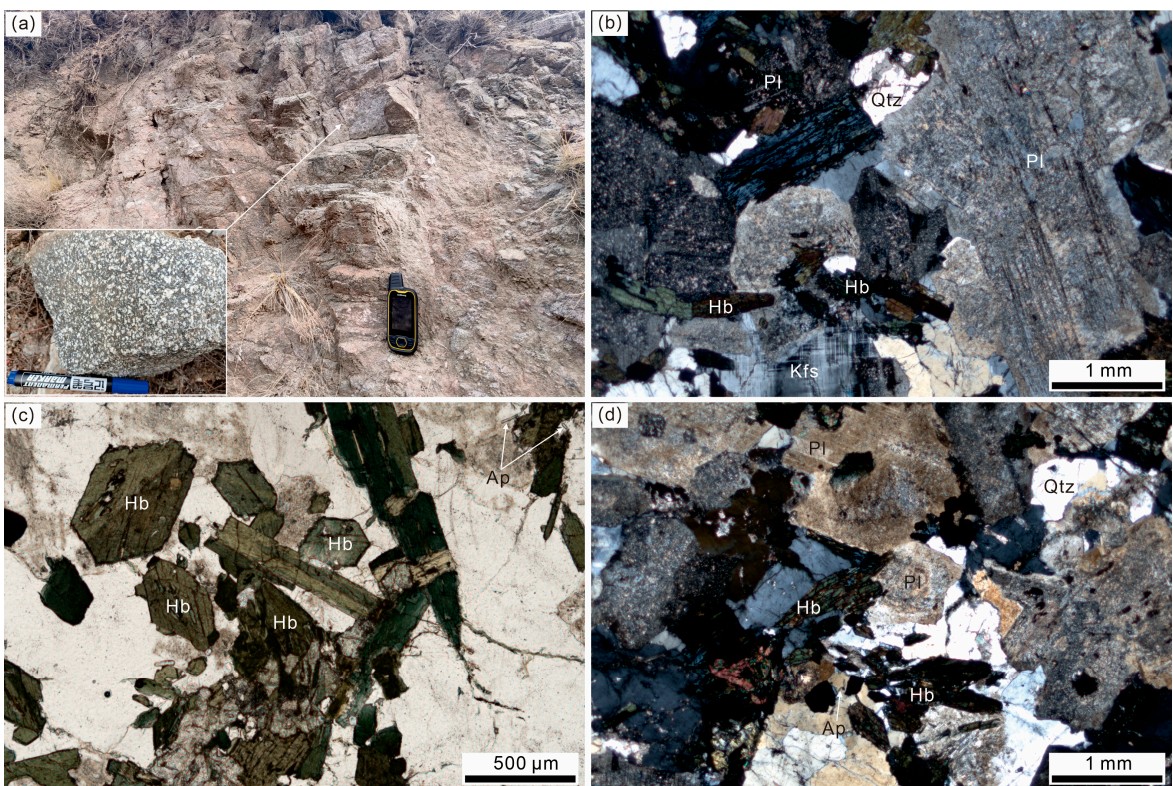

**Figure 2.** Field and microscope photographs of the Caowa dioritic intrusion in the Eastern NQOB. (**a**) Field photograph of the dioritic intrusion; (**b–d**) photomicrographs of the quartz diorites. Abbreviations: Hb = hornblende, Pl = plagioclase, Kfs = K-feldspar, Qtz = quartz, Ap = apatite.

## 3. Analytical Methods

### 3.1. Zircon U-Pb Dating and Hf Isotope Analyses

In this paper, zircon U-Pb dating and in situ Hf isotope analyses were carried out for Caowa quartz diorite from the Nanhuashan area in the eastern section of NQOB; the sampling geographic coordinates are 36°26′08″ N, 105°44′32″ E (Figure 1c). Density–magnetic techniques were used to separate zircon grains from a quartz diorite sample (CW-6) at the laboratory of the Hebei Regional Geological Survey Institute. Subsequently, zircon grains were polished until core areas were exposed, and pictures of cathodoluminescence (CL) were taken. Zircon U-Pb dating and trace element analyses were conducted simultaneously by LA-ICP-MS at the Wuhan SampleSolution Analytical Technology Co., Ltd. (Wuhan, China). Laser sampling was performed using a GeolasPro laser ablation system and a MicroLas optical system with a laser spot size of 32 μm and laser frequency of 8 Hz. An Agilent 7900 ICP-MS instrument was used to obtain ion-signal intensities [33]. Zircon 91,500, NIST 610, and GJ-1 were used as external standards for U-Pb dating and trace element calibration. Each analysis incorporated a background acquisition of approximately 20–30 s followed by 50 s of data acquisition from the sample. Detailed operating techniques and methods have been given by Zong et al. (2017) [34]. The software ICPMSDataCal was used for off-line selection, time-drift correction, and quantitative calibration for U-Pb dating and trace element analysis [35]. Concordia diagrams and weighted mean calculations were made using Isoplot [36].

In situ Hf isotope analyses of zircons were conducted using a Neptune Plus MC-ICP-MS (Thermo Fisher Scientific, Karlsruhe, Germany) in combination with an excimer ArF laser ablation system (Geolas HD) that was hosted at the Wuhan SampleSolution Analytical Technology Co., Ltd. During the analyses, a spot size of 32 μm and energy density of ~7.0 J cm$^{-2}$ were used. Detailed operating techniques and methods are the same as described by Hu et al. (2012) [37]. During analyses, three international zircon

standards (Plešovice, 91,500, and GJ-1) were analyzed simultaneously with the actual samples. The Hf isotope compositions of Plešovice, 91,500, and GJ-1 are 0.282478 ± 0.000008, 0.282300 ± 0.000011, and 0.282009 ± 0.000010, respectively [38].

### 3.2. Whole-Rock Major and Trace Element Analyses

Unaltered whole-rock samples were chosen and crushed to 60 mesh in a corundum jaw crusher, and about 60 g was powered in an agate ring mill to less than 200 mesh. Major and trace element analyses of the fresh whole-rock samples were carried out at Wuhan SampleSolution Analytical Technology Co., Ltd. Major element analyses were performed by an X-ray fluorescence spectrometer (XRF; Primus II, Rigaku, Austin, TX, USA), and the relative standard deviation was less than 2%. The samples for trace element analyses were digested by HF + HNO$_3$ solution in Teflon bombs at 195 °C for >24 h and then analyzed on an Agilent 7700e ICP-MS.

### 3.3. Whole-Rock Sr-Nd Isotope Analyses

The analyses of the whole-rock Sr-Nd isotope were conducted at Wuhan SampleSolution Analytical Technology Co., Ltd. by using a Neptune Plus MC-ICP-MS instrument. The Sr and Nd fractions were eluted using 2.5 and 0.3 M HCl, respectively. Isotopes were separated by conventional cation exchange. The mass fractionation correction for Sr and Nd isotopic ratios were normalized to $^{86}Sr/^{88}Sr$ = 0.1194 and $^{146}Nd/^{144}Nd$ = 0.7219, respectively. International standards (NBS987 and GSB) were used as bracketing standards to monitor the instrument drift during the analysis of Sr and Nd isotopes, respectively. Repeated analysis for NBS987 gave an average $^{87}Sr/^{86}Sr$ = 0.710242 ± 14 (2σ). Repeated analysis for GSB gave an average $^{143}Nd/^{144}Nd$ = 0.512440 ± 1 (2σ). The precision for $^{87}Rb/^{86}Sr$ was better than 1%, and the error in the $^{147}Sm/^{144}Nd$ was < 0.5%.

## 4. Results

### 4.1. Zircon U-Pb Geochronology

The zircon U-Pb isotopic data and trace element results of the Caowa dioritic intrusion are presented in Tables 1 and 2, respectively. The zircon CL images and concordia diagrams are shown in Figure 3.

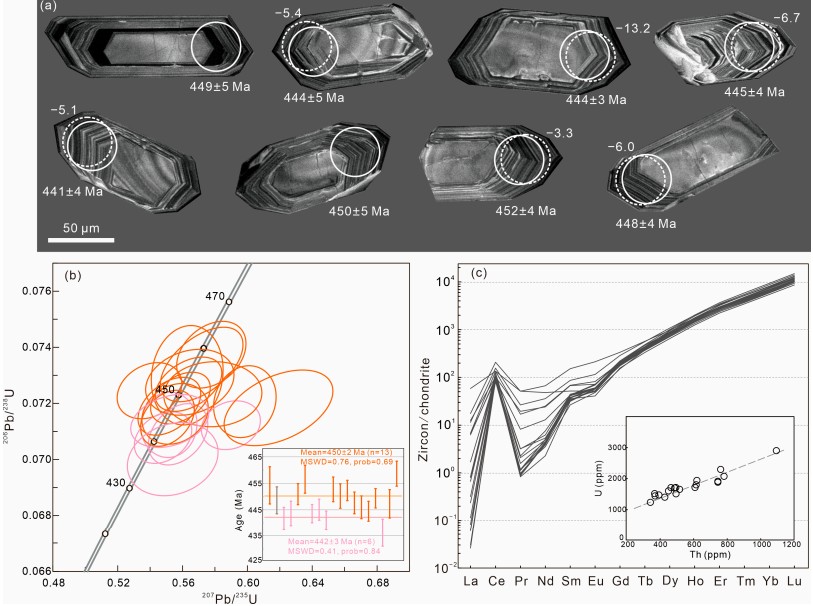

**Figure 3.** Cathodoluminescence images (**a**), LA-ICP-MS U-Pb concordia diagram (**b**), and chondrite-normalized REE patterns (**c**) of representative zircons from Caowa quartz diorite (**a**). Solid and dashed circles indicate the location of U-Pb analysis and Hf analysis, respectively.

**Table 1.** LA-ICP-MS zircon U-Pb dating results of the Caowa quartz diorite.

| Spot No. | Contents (ppm) | | Th/U | Isotopic Ratios | | | | | | | | Isotopic Ages (Ma) | | | | | |
|---|---|---|---|---|---|---|---|---|---|---|---|---|---|---|---|---|---|
| | $^{232}$Th | $^{238}$U | | $^{207}$Pb/$^{206}$Pb | 1σ | $^{207}$Pb/$^{235}$U | 1σ | $^{206}$Pb/$^{238}$U | 1σ | $^{208}$Pb/$^{232}$Th | 1σ | $^{207}$Pb/$^{206}$Pb | 1σ | $^{207}$Pb/$^{235}$U | 1σ | $^{206}$Pb/$^{238}$U | 1σ |
| CW-6-01 | 744 | 1889 | 0.39 | 0.0546 | 0.0018 | 0.5629 | 0.0146 | 0.0730 | 0.0012 | 0.0224 | 0.0005 | 394 | 79 | 453 | 10 | 454 | 7 |
| CW-6-02 | 1093 | 2859 | 0.38 | 0.0602 | 0.0013 | 0.5984 | 0.0127 | 0.0721 | 0.0008 | 0.0247 | 0.0006 | 613 | 46 | 476 | 8 | 449 | 5 |
| CW-6-03 | 781 | 2046 | 0.38 | 0.0559 | 0.0013 | 0.5572 | 0.0121 | 0.0709 | 0.0007 | 0.0215 | 0.0004 | 456 | 52 | 450 | 8 | 442 | 4 |
| CW-6-04 | 744 | 1860 | 0.40 | 0.0559 | 0.0013 | 0.5516 | 0.0114 | 0.0712 | 0.0008 | 0.0219 | 0.0004 | 456 | 52 | 446 | 7 | 444 | 5 |
| CW-6-05 | 370 | 1424 | 0.26 | 0.0538 | 0.0013 | 0.5396 | 0.0132 | 0.0725 | 0.0007 | 0.0213 | 0.0004 | 365 | 54 | 438 | 9 | 451 | 4 |
| CW-6-06 | 619 | 1916 | 0.32 | 0.0561 | 0.0013 | 0.5718 | 0.0168 | 0.0734 | 0.0009 | 0.0225 | 0.0004 | 457 | 52 | 459 | 11 | 457 | 5 |
| CW-6-07 | 368 | 1498 | 0.25 | 0.0599 | 0.0012 | 0.5898 | 0.0120 | 0.0712 | 0.0006 | 0.0251 | 0.0004 | 611 | 38 | 471 | 8 | 444 | 3 |
| CW-6-08 | 465 | 1687 | 0.28 | 0.0557 | 0.0010 | 0.5507 | 0.0110 | 0.0715 | 0.0007 | 0.0218 | 0.0004 | 443 | 43 | 445 | 7 | 445 | 4 |
| CW-6-09 | 342 | 1222 | 0.28 | 0.0560 | 0.0012 | 0.5480 | 0.0117 | 0.0708 | 0.0006 | 0.0220 | 0.0004 | 450 | 46 | 444 | 8 | 441 | 4 |
| CW-6-10 | 516 | 1633 | 0.32 | 0.0562 | 0.0018 | 0.5645 | 0.0180 | 0.0728 | 0.0008 | 0.0228 | 0.0005 | 457 | 72 | 454 | 12 | 453 | 5 |
| CW-6-11 | 393 | 1455 | 0.27 | 0.0577 | 0.0020 | 0.5787 | 0.0200 | 0.0723 | 0.0008 | 0.0272 | 0.0007 | 520 | 74 | 464 | 13 | 450 | 5 |
| CW-6-12 | 761 | 2264 | 0.34 | 0.0559 | 0.0013 | 0.5637 | 0.0135 | 0.0727 | 0.0006 | 0.0212 | 0.0004 | 456 | 50 | 454 | 9 | 452 | 4 |
| CW-6-13 | 494 | 1690 | 0.29 | 0.0553 | 0.0012 | 0.5519 | 0.0128 | 0.0719 | 0.0007 | 0.0230 | 0.0005 | 433 | 48 | 446 | 8 | 448 | 4 |
| CW-6-14 | 428 | 1386 | 0.31 | 0.0561 | 0.0014 | 0.5554 | 0.0135 | 0.0716 | 0.0007 | 0.0221 | 0.0005 | 457 | 21 | 449 | 9 | 446 | 4 |
| CW-6-15 | 615 | 1777 | 0.35 | 0.0556 | 0.0014 | 0.5521 | 0.0159 | 0.0714 | 0.0006 | 0.0226 | 0.0005 | 435 | 56 | 446 | 10 | 444 | 4 |
| CW-6-16 | 451 | 1584 | 0.28 | 0.0558 | 0.0012 | 0.5587 | 0.0122 | 0.0722 | 0.0006 | 0.0216 | 0.0004 | 443 | 51 | 451 | 8 | 449 | 4 |
| CW-6-17 | 495 | 1491 | 0.33 | 0.0571 | 0.0019 | 0.5540 | 0.0185 | 0.0700 | 0.0009 | 0.0233 | 0.0006 | 498 | 74 | 448 | 12 | 436 | 5 |
| CW-6-18 | 608 | 1693 | 0.36 | 0.0621 | 0.0020 | 0.6192 | 0.0221 | 0.0718 | 0.0009 | 0.0242 | 0.0012 | 676 | 69 | 489 | 14 | 447 | 5 |
| CW-6-19 | 487 | 1678 | 0.29 | 0.0563 | 0.0014 | 0.5778 | 0.0141 | 0.0737 | 0.0008 | 0.0233 | 0.0006 | 465 | 54 | 463 | 9 | 459 | 5 |

**Table 2.** Zircon trace element data of the Caowa quartz diorite.

| Spot No. | La | Ce | Pr | Nd | Sm | Eu | Gd | Tb | Dy | Ho | Er | Tm | Yb | Lu | Eu/Eu* | Hf | Ta | Y | Ti | Nb |
|---|---|---|---|---|---|---|---|---|---|---|---|---|---|---|---|---|---|---|---|---|
| CW-6-01 | 0.16 | 65.71 | 0.30 | 2.50 | 5.64 | 3.43 | 41.79 | 15.29 | 207 | 88.11 | 442 | 106.56 | 1095 | 257 | 0.49 | 28,295 | 6.09 | 2803 | 13.32 | 18.18 |
| CW-6-02 | 2.71 | 95.28 | 2.51 | 16.76 | 13.22 | 6.67 | 59.00 | 21.00 | 273 | 113.41 | 556 | 132.00 | 1356 | 306 | 0.62 | 27,149 | 8.94 | 3634 | 22.20 | 33.09 |
| CW-6-03 | 0.01 | 66.60 | 0.08 | 1.92 | 6.56 | 3.44 | 42.17 | 16.44 | 217 | 91.56 | 472 | 113.50 | 1184 | 277 | 0.48 | 28,530 | 6.55 | 2957 | 11.24 | 19.62 |
| CW-6-04 | 0.20 | 76.26 | 0.21 | 3.53 | 6.55 | 4.17 | 44.84 | 17.58 | 227 | 97.89 | 498 | 118.97 | 1248 | 292 | 0.55 | 27,469 | 6.58 | 3153 | 12.14 | 22.23 |
| CW-6-05 | 0.02 | 50.15 | 0.11 | 1.65 | 4.07 | 2.80 | 30.96 | 13.94 | 189 | 80.95 | 418 | 102.28 | 1088 | 261 | 0.54 | 27,647 | 5.66 | 2607 | 9.11 | 15.53 |
| CW-6-06 | 13.94 | 82.46 | 4.87 | 22.53 | 7.96 | 3.02 | 38.18 | 13.71 | 181 | 76.11 | 395 | 94.46 | 1003 | 239 | 0.44 | 29,534 | 5.88 | 2473 | 7.24 | 17.51 |
| CW-6-07 | 2.96 | 62.56 | 1.85 | 11.27 | 10.34 | 4.18 | 41.91 | 14.74 | 210 | 92.24 | 490 | 122.45 | 1296 | 311 | 0.53 | 29,319 | 6.04 | 3019 | 13.23 | 18.31 |
| CW-6-08 | 0.00 | 60.01 | 0.09 | 2.03 | 6.29 | 3.34 | 44.66 | 16.44 | 239 | 108.89 | 573 | 139.26 | 1488 | 355 | 0.45 | 28,350 | 6.75 | 3530 | 10.76 | 23.00 |
| CW-6-09 | 0.01 | 51.36 | 0.11 | 2.37 | 5.80 | 3.42 | 40.01 | 15.27 | 228 | 100.04 | 538 | 131.31 | 1376 | 327 | 0.51 | 27,151 | 5.18 | 3246 | 11.26 | 18.19 |
| CW-6-10 | 0.06 | 55.93 | 0.15 | 2.10 | 4.98 | 3.37 | 36.84 | 14.65 | 201 | 85.35 | 452 | 110.00 | 1156 | 273 | 0.55 | 28,330 | 6.46 | 2800 | 11.83 | 19.52 |
| CW-6-11 | 1.61 | 82.48 | 1.61 | 11.68 | 10.50 | 4.74 | 42.03 | 14.21 | 192 | 83.62 | 420 | 98.51 | 1029 | 252 | 0.60 | 29,354 | 4.25 | 2626 | 22.40 | 13.69 |
| CW-6-12 | 0.08 | 60.44 | 0.08 | 1.06 | 4.83 | 3.30 | 35.79 | 13.82 | 191 | 83.22 | 420 | 102.55 | 1069 | 252 | 0.55 | 29,612 | 7.33 | 2661 | 9.38 | 19.93 |
| CW-6-13 | 0.01 | 62.43 | 0.11 | 2.33 | 6.54 | 3.99 | 45.41 | 18.07 | 254 | 114.76 | 605 | 150.52 | 1592 | 380 | 0.52 | 27,316 | 6.95 | 3712 | 13.38 | 22.66 |
| CW-6-14 | 1.56 | 56.25 | 0.78 | 6.05 | 6.50 | 3.42 | 40.46 | 15.75 | 209 | 92.13 | 485 | 117.96 | 1243 | 297 | 0.49 | 27,661 | 5.72 | 2963 | 30.72 | 17.65 |
| CW-6-15 | 0.40 | 62.18 | 0.30 | 2.90 | 6.80 | 3.51 | 42.94 | 15.20 | 214 | 91.05 | 469 | 113.20 | 1174 | 280 | 0.48 | 27,915 | 6.77 | 2907 | 10.23 | 19.48 |
| CW-6-16 | 0.01 | 53.87 | 0.11 | 2.03 | 5.84 | 2.90 | 38.17 | 15.39 | 211 | 94.62 | 492 | 120.89 | 1268 | 304 | 0.45 | 28,605 | 6.03 | 3029 | 11.04 | 20.50 |
| CW-6-17 | 0.19 | 57.12 | 0.20 | 2.29 | 5.43 | 2.38 | 32.30 | 12.46 | 165 | 70.94 | 370 | 88.25 | 928 | 220 | 0.43 | 28,248 | 5.33 | 2278 | 21.60 | 13.96 |
| CW-6-18 | 4.17 | 127.01 | 4.87 | 31.21 | 23.26 | 12.35 | 68.83 | 21.36 | 245 | 98.51 | 473 | 107.22 | 1144 | 262 | 0.87 | 26,455 | 6.09 | 3042 | 15.51 | 18.31 |
| CW-6-19 | 0.03 | 55.17 | 0.09 | 1.80 | 5.20 | 3.05 | 39.80 | 15.83 | 224 | 99.35 | 531 | 132.58 | 1388 | 334 | 0.46 | 28,114 | 6.50 | 3256 | 10.68 | 21.27 |

The zircon grains in the Caowa quartz diorite sample CW-6 are colorless to faint yellow, transparent, and with euhedral morphology. They have a size range of 50–150 μm, with a length/width ratio of 2:1 to 3:1, and show broad oscillatory growth zoning or are homogeneous in the CL images (Figure 3a). The analyzed zircons have U and Th contents of 1222–2859 and 342–1093 ppm, and show high Th/U ratios ranging from 0.25 to 0.40. They display enrichment in HREEs and depletion in LREEs, with obvious positive Ce anomalies, weak negative Eu anomalies, and positive correlation between Th and U contents (Figure 3c), indicating their magmatic origin [39,40]. We analyzed 19 spots on 19 zircons, with $^{206}Pb/^{238}U$ ages ranging from $436 \pm 5$ to $459 \pm 5$ Ma. Among them, 6 spots displayed younger $^{206}Pb/^{238}U$ ages between $436 \pm 5$ and $444 \pm 4$ Ma, with a weighted mean of $442 \pm 3$ Ma (MSWD = 0.41), and the remaining 13 spots yielded older $^{206}Pb/^{238}U$ ages ranging from $445 \pm 4$ to $459 \pm 5$ Ma, with a weighted mean of $450 \pm 2$ Ma (MSWD = 0.76) (Figure 3b). We consider that the age of $450 \pm 2$ Ma represents the emplacement age of the Caowa dioritic intrusion.

### 4.2. Whole-Rock Major and Trace Elements

Major and trace element data of the Caowa quartz diorites are listed in Table 3. Based on petrographic examination, all samples are not significantly affected by hydrothermal alteration, and their low loss on ignition (LOI) contents (1.36–2.45 wt.%, Table 3) further indicate that the samples are basically fresh.

**Table 3.** Major (wt.%), trace element (ppm) and Sr-Nd isotopic compositions of the Caowa quartz diorites.

| Sample | CW-1 | CW-2 | CW-3 | CW-4 | CW-5 | CW-6 |
|---|---|---|---|---|---|---|
| $SiO_2$ | 62.80 | 62.87 | 59.75 | 58.34 | 57.53 | 61.97 |
| $TiO_2$ | 0.61 | 0.59 | 0.74 | 0.77 | 0.90 | 0.63 |
| $Al_2O_3$ | 16.32 | 16.85 | 16.99 | 16.66 | 17.16 | 16.90 |
| $Fe_2O_3^T$ | 5.39 | 5.22 | 6.58 | 6.63 | 7.88 | 5.54 |
| MnO | 0.13 | 0.12 | 0.14 | 0.14 | 0.17 | 0.12 |
| MgO | 2.13 | 2.10 | 2.43 | 2.63 | 3.01 | 2.08 |
| CaO | 3.47 | 3.71 | 4.65 | 5.46 | 5.26 | 3.81 |
| $Na_2O$ | 3.91 | 3.95 | 3.78 | 3.71 | 3.65 | 3.86 |
| $K_2O$ | 2.95 | 2.93 | 2.78 | 2.94 | 2.50 | 2.82 |
| $P_2O_5$ | 0.19 | 0.19 | 0.23 | 0.23 | 0.26 | 0.20 |
| LOI | 2.13 | 1.36 | 1.42 | 2.453 | 1.43 | 1.62 |
| Total | 100.03 | 99.90 | 99.48 | 99.97 | 99.74 | 99.56 |
| $Mg^\#$ | 44 | 44 | 42 | 44 | 43 | 43 |
| A/CNK | 1.02 | 1.02 | 0.96 | 0.87 | 0.94 | 1.03 |
| $Na_2O + K_2O$ | 6.86 | 6.89 | 6.55 | 6.65 | 6.16 | 6.68 |
| Sc | 9.22 | 8.41 | 11.84 | 11.76 | 15.11 | 9.24 |
| V | 78.3 | 80.2 | 105.7 | 106.7 | 130.8 | 83.7 |
| Cr | 4.81 | 5.21 | 5.69 | 5.45 | 6.43 | 4.58 |
| Co | 8.92 | 9.08 | 11.92 | 11.89 | 14.65 | 9.57 |
| Ni | 3.29 | 3.46 | 4.53 | 4.61 | 5.12 | 3.71 |
| Cu | 9.87 | 8.70 | 11.77 | 14.24 | 23.66 | 4.63 |
| Zn | 70.0 | 69.0 | 80.5 | 78.6 | 93.0 | 67.9 |
| Rb | 92.1 | 90.0 | 81.6 | 80.2 | 76.6 | 72.9 |
| Sr | 561 | 612 | 623 | 601 | 646 | 646 |
| Y | 25.07 | 22.93 | 28.90 | 28.01 | 33.24 | 23.98 |
| Zr | 193.6 | 180.5 | 218.9 | 206.7 | 228.8 | 193.5 |
| Nb | 12.53 | 13.02 | 13.42 | 13.84 | 13.05 | 12.20 |
| Ba | 1165 | 916 | 860 | 1151 | 822 | 902 |
| La | 60.86 | 40.90 | 27.52 | 18.27 | 57.29 | 44.74 |
| Ce | 112.96 | 77.92 | 53.83 | 37.62 | 111.28 | 85.59 |
| Pr | 12.26 | 8.69 | 6.64 | 5.05 | 12.74 | 9.50 |
| Nd | 41.73 | 30.58 | 25.94 | 21.20 | 45.27 | 33.69 |
| Sm | 7.06 | 5.42 | 5.92 | 5.23 | 8.40 | 6.20 |

**Table 3.** *Cont.*

| Sample | CW-1 | CW-2 | CW-3 | CW-4 | CW-5 | CW-6 |
|---|---|---|---|---|---|---|
| Eu | 1.58 | 1.36 | 1.60 | 1.57 | 1.96 | 1.54 |
| Gd | 5.04 | 4.30 | 5.20 | 4.96 | 6.60 | 4.66 |
| Tb | 0.74 | 0.64 | 0.81 | 0.79 | 1.00 | 0.70 |
| Dy | 4.30 | 3.70 | 4.92 | 4.67 | 5.89 | 3.98 |
| Ho | 0.86 | 0.79 | 0.96 | 1.00 | 1.17 | 0.83 |
| Er | 2.59 | 2.32 | 2.93 | 2.87 | 3.32 | 2.50 |
| Tm | 0.37 | 0.34 | 0.44 | 0.42 | 0.48 | 0.37 |
| Yb | 2.50 | 2.37 | 2.86 | 2.87 | 3.17 | 2.38 |
| Lu | 0.39 | 0.36 | 0.44 | 0.43 | 0.49 | 0.37 |
| Hf | 5.02 | 4.60 | 5.44 | 5.11 | 5.71 | 4.76 |
| Ta | 0.76 | 0.90 | 0.80 | 0.81 | 0.65 | 0.73 |
| Pb | 21.10 | 18.48 | 15.75 | 18.52 | 17.27 | 19.10 |
| Th | 15.53 | 10.99 | 6.12 | 4.29 | 14.31 | 11.99 |
| U | 1.68 | 1.39 | 1.57 | 1.68 | 2.17 | 1.79 |
| REE | 253.22 | 179.69 | 140.01 | 106.95 | 259.06 | 197.05 |
| Eu/Eu* | 0.77 | 0.83 | 0.86 | 0.93 | 0.78 | 0.84 |
| Sr/Y | 22.38 | 26.68 | 21.57 | 21.44 | 19.44 | 26.93 |
| Nb/Ta | 16.38 | 14.46 | 16.70 | 16.99 | 20.22 | 16.70 |
| $(La/Yb)_N$ | 17.47 | 12.36 | 6.91 | 4.56 | 12.95 | 13.47 |
| $(Gd/Yb)_N$ | 1.67 | 1.50 | 1.51 | 1.43 | 1.72 | 1.62 |
| $^{87}Rb/^{86}Sr$ | | 0.4258 | | 0.3863 | | 0.3267 |
| $^{87}Sr/^{86}Sr$ | | 0.711129 | | 0.710716 | | 0.710747 |
| $(^{87}Sr/^{86}Sr)_i$ | | 0.708338 | | 0.708184 | | 0.708606 |
| $^{147}Sm/^{144}Nd$ | | 0.1071 | | 0.1493 | | 0.1113 |
| $^{143}Nd/^{144}Nd$ | | 0.512124 | | 0.512243 | | 0.512119 |
| $\varepsilon_{Nd}(t)$ | | −4.8 | | −4.9 | | −5.1 |
| $T_{DM2}$ (Ma) | | 1579 | | 1591 | | 1607 |

The Caowa quartz diorites have variable $SiO_2$ contents (57.53–62.87 wt.%) and are of intermediate diorite or monzonite compositions (Figure 4a). They have high-K calc-alkaline and metaluminous to weak peraluminous features, as indicated by their high total alkalis ($Na_2O + K_2O$ = 6.16–6.89 wt.%), and low A/CNK values (0.87–1.03) (Figure 4b–d). They have high $Fe_2O_3^T$ (5.22–7.88 wt.%), CaO (3.47–5.46 wt.%) contents, and moderate MgO (2.08–3.01 wt.%) and $Mg^\#$ (42–44) contents. The Caowa quartz diorites have higher contents of REEs, ranging from 106.95 to 259.06 ppm (Table 3). The samples have high La (18.27–60.86 ppm) and low Yb (2.37–3.17 ppm) contents, and the $(La/Yb)_N$ ratios range from 4.56 to 17.47 (Table 3). Chondrite-normalized REE patterns display that all samples are enriched in LREEs and relatively depleted in HREEs, with no significant negative Eu anomalies (Eu/Eu* = 0.77–0.93; Figure 5a). The Caowa quartz diorites display extremely high Ba (up to 1165 ppm) and Sr (561 to 646 ppm) contents, with obvious enrichment in LILEs (Th, U, and K) and depletion in HFSEs (Nb, Ta, Ti, and P; Figure 5b).

*4.3. Whole-Rock Sr-Nd Isotopes*

Whole-rock Sr-Nd isotopic data for selected samples are presented in Table 3 and shown in Figure 6. The initial Sr-Nd isotopic ratios were calculated at 450 Ma based on the zircon U-Pb age. The Caowa quartz diorites show heterogeneous Sr-Nd isotope compositions characterized by slightly high $(^{87}Sr/^{86}Sr)_i$ ratios (0.70818–0.70860), and lower $\varepsilon_{Nd}(t)$ values (−5.1 to −4.9), with two-stage Nd model ages ($T_{DM2}$) of 1579–1607 Ma.

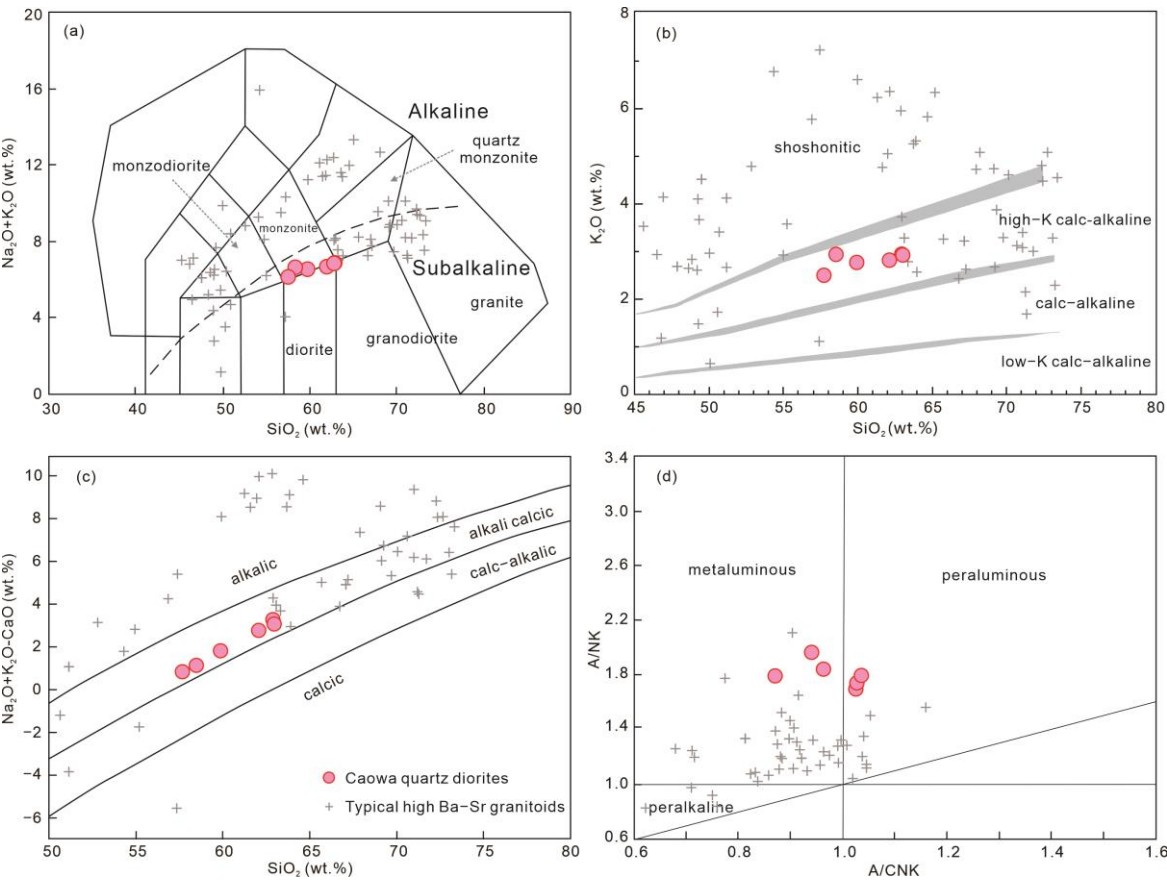

**Figure 4.** Geochemical classification of the Caowa quartz diorites. (**a**) Total alkali vs. silica (TAS) diagram (after [41]); (**b**) $K_2O$ vs. $SiO_2$ diagram (after [42]); (**c**) ($Na_2O + K_2O − CaO$) vs. $SiO_2$ diagram (after [43]); and (**d**) A/NK [molar ratio $Al_2O_3/(Na_2O + K_2O)$] vs. A/CNK [molar ratio $Al_2O_3/(CaO + Na_2O + K_2O)$] diagram (after [44]). The data for typical high Ba-Sr granitoids are from [15,19].

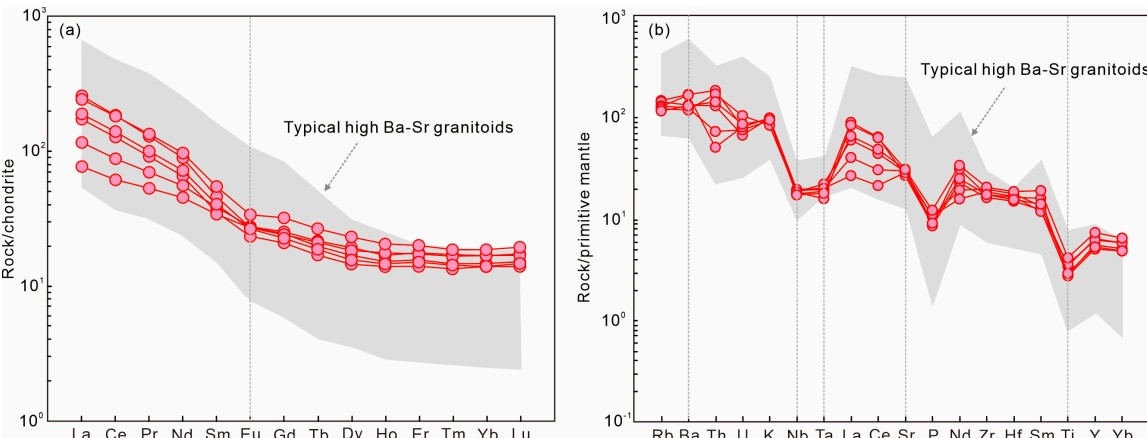

**Figure 5.** Chondrite-normalized REE patterns (**a**) and primitive mantle-normalized trace element patterns (**b**) for the Caowa quartz diorites. Normalizing values are from [45]. The compositions for typical high Ba-Sr granitoids are from [15,19].

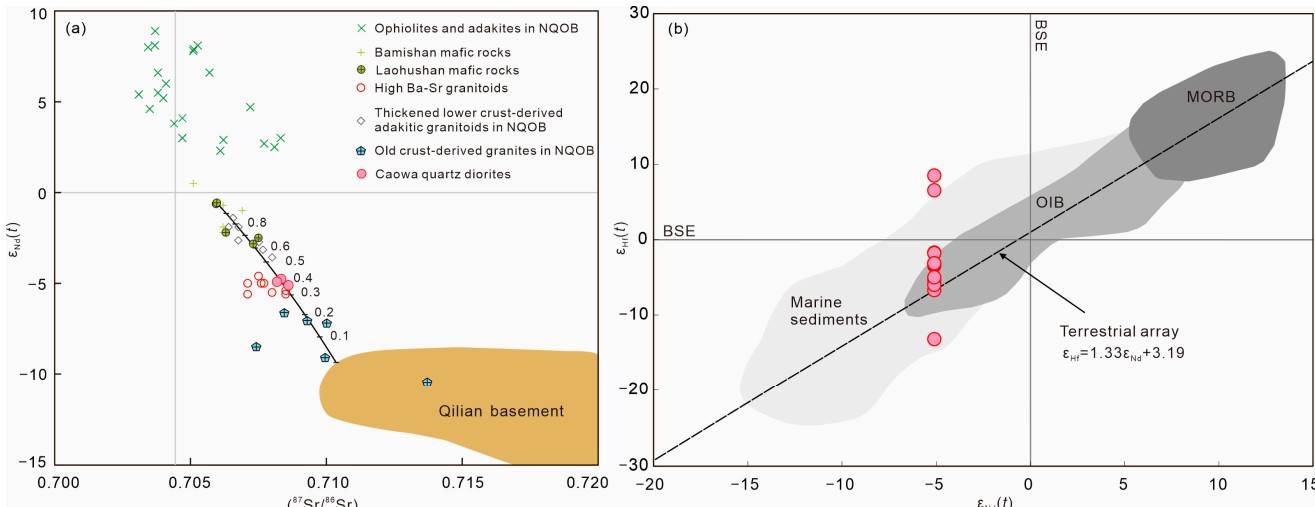

**Figure 6.** Initial $^{87}Sr/^{86}Sr$ vs. $\varepsilon_{Nd}(t)$ (**a**) and zircon $\varepsilon_{Hf}(t)$ vs. $\varepsilon_{Nd}(t)$ diagrams (**b**) for the Caowa quartz diorites. Data sources: depleted mantle-derived ophiolites and adakites in the NQOB are from [46–48]; enriched mantle-derived Bamishan and Laohushan mafic rocks and high Ba-Sr granitoids in the NQOB are from [49]; thickened lower crust-derived adakitic rocks and old crust-derived granites in the NQOB are from [9,10,22,50]; Qilian basement from [51]. Fields of MORB, OIB and marine sediments, and Terrestrial array after [52].

### 4.4. Zircon Hf Isotopes

In situ Hf isotope analyses of zircons from the Caowa quartz diorite are listed in Table 4 and shown in Figure 7. The Caowa quartz diorite displays variable $^{176}Hf/^{177}Hf$ ratios (0.282135–0.282753) and $\varepsilon_{Hf}(t)$ values (−13.2 to +8.5), with two-stage Hf model ages ($T_{DM}{}^{c}$) of 830–2029 Ma. The wide ranges of $\varepsilon_{Hf}(t)$ values are similar to the granitoids of crust–mantle mixed source in the NQOB (Figure 7).

**Table 4.** Zircon Hf isotopic compositions of the Caowa quartz diorites.

| Spot No. | Age (Ma) | $^{176}Yb/^{177}Hf$ | $^{176}Lu/^{177}Hf$ | $^{176}Hf/^{177}Hf$ | 1σ | $\varepsilon_{Hf}(0)$ | $\varepsilon_{Hf}(t)$ | $T_{DM}$ (Ma) | $T_{DM}{}^{c}$ (Ma) | $f_{Lu/Hf}$ |
|---|---|---|---|---|---|---|---|---|---|---|
| CW-6-01 | 454 | 0.047583 | 0.001043 | 0.282401 | 0.000028 | −13.1 | −3.5 | 1204 | 1499 | −0.97 |
| CW-6-03 | 442 | 0.060913 | 0.001249 | 0.282459 | 0.000102 | −11.1 | −1.7 | 1128 | 1393 | −0.96 |
| CW-6-04 | 444 | 0.053286 | 0.001496 | 0.282355 | 0.000036 | −14.7 | −5.4 | 1283 | 1599 | −0.95 |
| CW-6-05 | 451 | 0.041103 | 0.000944 | 0.282449 | 0.000048 | −11.4 | −1.8 | 1133 | 1404 | −0.97 |
| CW-6-06 | 457 | 0.051349 | 0.001116 | 0.282409 | 0.000073 | −12.8 | −3.1 | 1195 | 1483 | −0.97 |
| CW-6-07 | 444 | 0.074723 | 0.001580 | 0.282135 | 0.000366 | −22.5 | −13.2 | 1598 | 2029 | −0.95 |
| CW-6-08 | 445 | 0.067271 | 0.001497 | 0.282319 | 0.000104 | −16.0 | −6.7 | 1335 | 1670 | −0.95 |
| CW-6-09 | 441 | 0.043239 | 0.001363 | 0.282365 | 0.000032 | −14.4 | −5.1 | 1264 | 1578 | −0.96 |
| CW-6-10 | 453 | 0.053681 | 0.001471 | 0.282335 | 0.000066 | −15.4 | −5.9 | 1310 | 1634 | −0.96 |
| CW-6-12 | 452 | 0.067973 | 0.001496 | 0.282411 | 0.000057 | −12.8 | −3.3 | 1204 | 1487 | −0.95 |
| CW-6-13 | 448 | 0.062911 | 0.001953 | 0.282340 | 0.000028 | −15.3 | −6.0 | 1320 | 1634 | −0.94 |
| CW-6-14 | 446 | 0.079210 | 0.002148 | 0.282753 | 0.000114 | −0.7 | 8.5 | 730 | 830 | −0.94 |
| CW-6-15 | 444 | 0.057545 | 0.001758 | 0.282422 | 0.000041 | −12.4 | −3.1 | 1197 | 1473 | −0.95 |
| CW-6-16 | 449 | 0.059503 | 0.001403 | 0.282364 | 0.000085 | −14.4 | −5.0 | 1268 | 1579 | −0.96 |
| CW-6-19 | 459 | 0.061641 | 0.001196 | 0.282682 | 0.000099 | −3.2 | 6.6 | 812 | 948 | −0.96 |

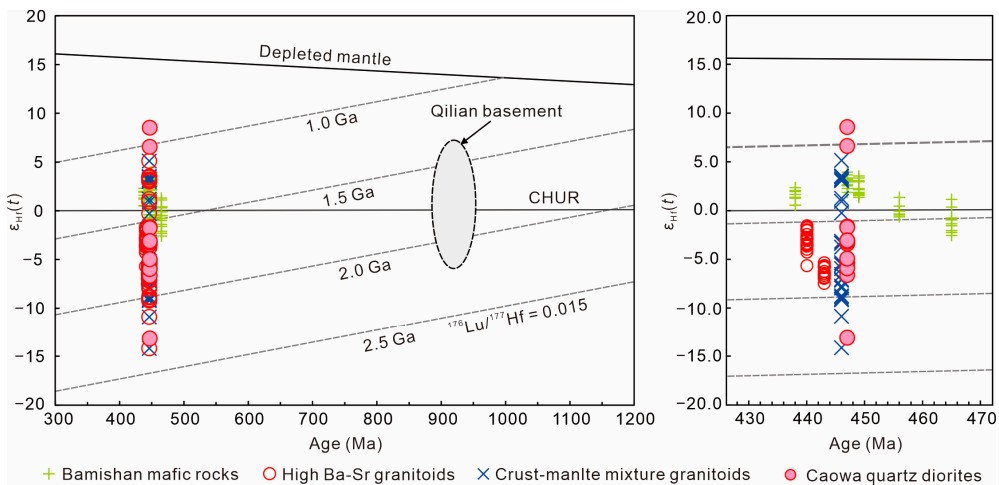

**Figure 7.** Zircon $\varepsilon_{Hf}(t)$ vs. age ($t$) diagram for the Caowa quartz diorites. All $\varepsilon_{Hf}(t)$ values were calculated at the ages given by the U-Pb data. Data sources: crust–mantle mixture granitoids in the NQOB are from [53,54]. See Figure 6 for other data sources.

## 5. Discussion

### 5.1. Petrogenesis

The major element data show that the Caowa quartz diorites are high-K calc-alkaline rocks and have slightly higher MgO content. The studied samples are enriched in LILEs (e.g., Ba, Th, and U) and LREEs and depleted in HFSEs (e.g., Nb, Ta, and Ti); these geochemical features suggest that they are similar to arc magmatic rocks in subduction zones [55]. In addition, the Caowa quartz diorites carry some geochemical characteristics of trace elements that distinguish them from traditional I-, S-, and A-type granitoids. In the Rb-Ba-Sr diagram (Figure 8), the samples plot in the field of high Ba-Sr granitoids. The samples show high Ba (822–1165 ppm) and Sr (561–646 ppm) contents and low Rb (72.9–92.1 ppm) and U (1.39–2.17 ppm) contents, together with enrichment in LREEs and depletion in HREEs, and negligible Eu anomalies (Figure 5), which are similar to the typical high Ba-Sr granitoids worldwide (e.g., Northern Scotland, Eastern China) [14,16,17,20]. Several petrogenetic models have been proposed for the high Ba-Sr granitoids: (1) partial melting of subducted ocean islands/ocean plateaus [14]; (2) partial melting of thickened mafic lower crust with or without mantle input [18,24,56]; (3) partial melting of enriched lithospheric mantle metasomatized by subduction-related fluids or melts [19,21,22]; (4) mixing of enriched mantle-derived mafic magmas with crustal felsic melts [16,17,25].

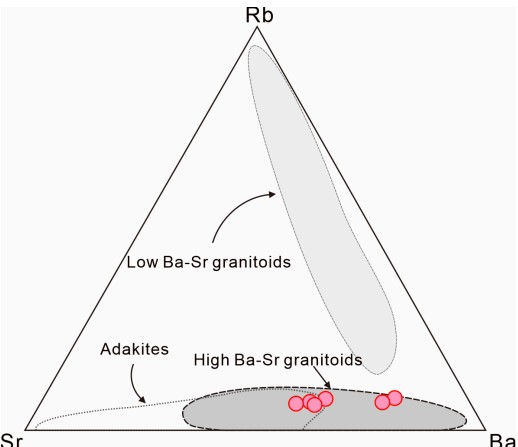

**Figure 8.** Rb-Ba-Sr ternary diagram for the Caowa quartz diorites, modified after [14].

Tarney and Jones (1994) first proposed that partial melting of subducted oceanic islands or plateaus can generate high Ba-Sr initial magmas, which can also explain their strong fractionated REE patterns and negligible Eu anomalies [14]. However, based on the findings from experimental petrology, partial melting of basaltic oceanic crust (MORB or OIB) usually produces Na-rich magmas ($Na_2O$ > 5 wt.%) [57], which is evidently inconsistent with the high-K calc-alkaline characteristics of the Caowa quartz diorites (Figure 4b). In addition, the Caowa quartz diorites have enriched Sr-Nd isotope compositions with initial $^{87}Sr/^{86}Sr$ ratios = 0.70818–0.70860 and $\varepsilon_{Nd}(t)$ = −5.1 to −4.9 that are obviously distinct from those of depleted mantle-derived Early Paleozoic ophiolites and adakites in NQOB [46–48] (Figure 6a). The partial melting of basaltic oceanic crust cannot generate the Caowa high Ba-Sr quartz diorites.

It is generally accepted that partial melting of thickened mafic lower continental crust is a key mechanism to produce high Ba-Sr granitoids [19,25,56], which is mainly due to their affinities with adakitic granitoids, such as: high alkali, Sr, and LREE contents; low Rb, Y, and HREE contents; and high Sr/Y and La/Yb ratios [20,58]. Although the rocks studied here display high Sr contents, enrichment in LREEs, and negligible Eu anomalies (some adakitic features), they have high Y (22.93–33.24 ppm) and Yb (2.37–3.17 ppm) contents and low Sr/Y (19–27) and La/Yb (4.56–17.47) ratios significantly different from typical adakites (Figure 9), and more evolved Sr-Nd isotope compositions than the adakitic granitoids that are proposed to have originated from partial melting of thickened lower crust in the eastern part of NQOB [9,10,13] (Figure 6a). Furthermore, magmas, derived from high-pressure partial melting of the thickened lower crust, generally possess extinct fractioned HREEs (e.g., Gd/Yb ratios > 8) [59]. All samples studied here display low Gd/Yb ratios (1.43–1.72) and flat HREE distribution patterns (Figure 5a), which indicates that there was no significant involvement of garnet during the magmatic generation. Consequently, the thickened crust model for the petrogenesis of the high Ba-Sr granitoids may not be applicable to the Caowa quartz diorites.

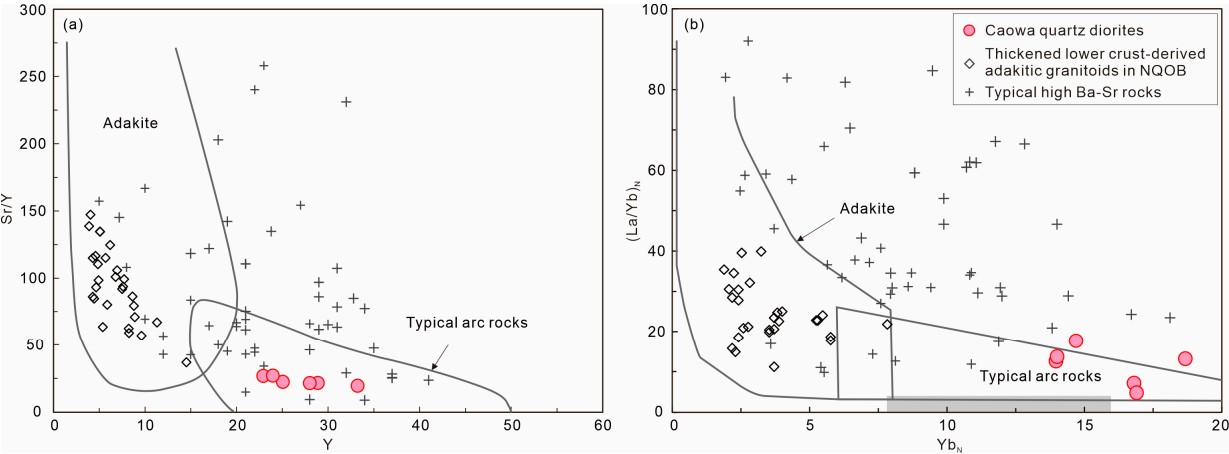

**Figure 9.** Discrimination diagrams of Sr/Y vs. Y (**a**) and $(La/Yb)_N$ vs. $Yb_N$ (**b**) for the Caowa quartz diorites, modified after [58]. See Figure 6 for the data sources.

Previous studies have shown that subducted oceanic slab and sediment-derived melts or fluids have a high capacity to carry significant amounts of Ba and Sr [60,61], and transfer of these elements would result in enrichment of the overlying lithospheric mantle through metasomatism. Thus, low-degree partial melting of enriched lithospheric mantle metasomatized by subduction-related fluids or melts can generate initial magmas with high Ba-Sr signatures [19,21,22]. The Caowa quartz diorites exhibit higher Nd (21.20–45.27 ppm) and Nb (12.20–13.84 ppm) contents and Nb/Ta (14.46–20.22) values than those derived from the continental crust (Nd = 11–27, average Nb/Ta = 11) [45], indicating the significant contribution of mantle components. Moreover, the Caowa quartz diorites have relatively enriched Sr-Nd-Hf isotopic compositions, which are consistent with those of the enriched

mantle-derived high Ba-Sr granitoids from the southern margin of Alxa block (Figures 6a and 7) [22]. As shown in plots of Ba/Th vs. Th/Zr and Rb/Y vs. Nb/Y (Figure 10), magma sources for the Caowa quartz diorites were probably related to the lithospheric mantle metasomatized by subduction-related fluids. This is also confirmed by the Nd-Hf isotope decoupling (Figure 6b), owing to the discrepant elemental behavior between Nd and Hf. Normally, Nd is much more mobile than Hf in subduction zones. It is difficult to cause Hf isotope enrichment when metasomatism occurs with the overlying lithospheric mantle [62,63], which is consistent with the occurrence of higher $\varepsilon_{Hf}(t)$ values (e.g., −3.5 to −1.7) than $\varepsilon_{Nd}(t)$ values (−5.1 to −4.8) for the Caowa dioritic intrusion. More importantly, the positive $\varepsilon_{Hf}(t)$ values (e.g., +6.6 and +8.5) likely indicate that there may also be a very small quantity of oceanic crust materials involved in the magma source of the Caowa quartz diorites. However, it should be noted that partial melting of enriched mantle model provides a certain degree of support for the high Ba-Sr signatures of Caowa quartz diorites, but the relatively low MgO (2.08–3.01 wt.%), Mg# (42–44), Cr (4.58–6.43 ppm), and Ni (3.29–5.12 ppm) contents of these rocks are distinct from the high-Mg diorites and clearly argue against a single, common mantle evolution by partial melting processes [64,65]. Since such a mechanism is not feasible here, we interpret that the magma source of these rocks may also have the addition of continental crustal components, and the extremely low zircon $\varepsilon_{Hf}(t)$ values (as low as −13.2) of the Caowa quartz diorite also manifest the additional ancient crust composition in the magma source.

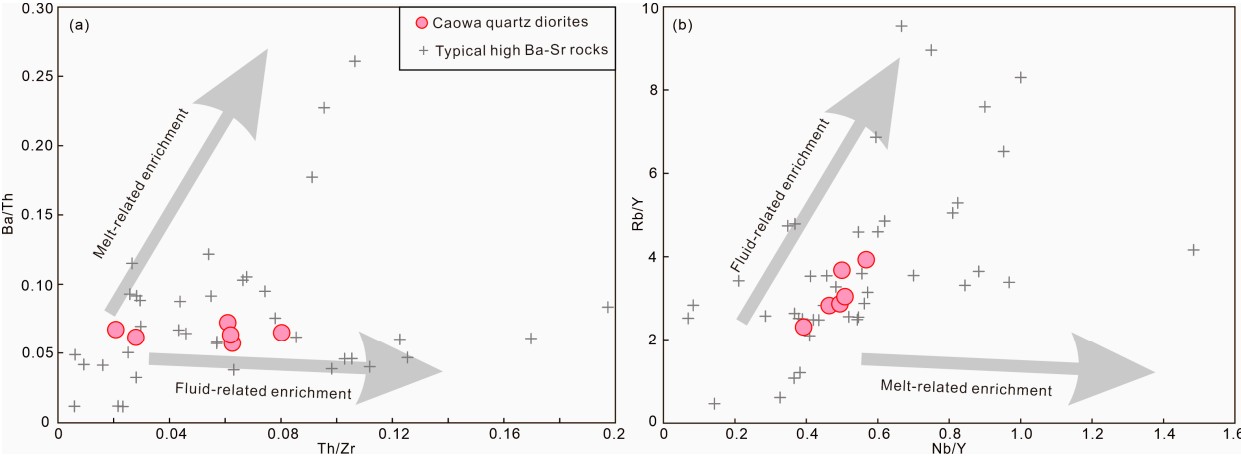

**Figure 10.** Ba/Th vs. Th/Zr (**a**) and Rb/Yb vs. Nb/Yb (**b**) diagrams for the Caowa quartz diorites, modified after [21].

Here, we propose that the studied Caowa high Ba-Sr quartz diorites probably generated through mixing of enriched lithospheric mantle-derived basaltic and crustal felsic magmas. Their moderate SiO$_2$ (57.53–62.87 wt.%) and MgO (2.08–3.01 wt.%) contents and Mg# values (42–44), as well as the wide ranges of zircon Hf isotopes, support the mechanism of crust–mantle interaction (Figure 7). The studied rocks exhibit more radiogenic Sr and Nd isotopes when compared to the enriched mantle-derived mafic rocks from NQOB, and are also different from those of the Precambrian metamorphic basement and associated granites (Figure 6a) [50,51]. A simple isotope model was adopted to evaluate the possible mixing proportion of mantle and crustal components, and it is suggested that the Caowa high Ba-Sr quartz diorites might be products of mixing of 40% enriched mantle and 60% ancient crustal melts (Figure 6a). Moreover, the Caowa quartz diorites are associated with the contemporary mafic rocks, dioritic enclaves, and felsic rocks in the surrounding areas, such as Laohushan-Quwushan Mountain (Figure 1b), which were considered to be derived from partial melting of the metasomatized enriched lithosphere mantle, magma mixing, and partial melting of lower continental crust, respectively [11,13,66,67]. Successive variation in major elemental compositions between them (Figure 11) further substantiates

a petrogenetic model of crust–mantle interaction [68,69]. Such a crust–mantle interaction mechanism can be further supported by the hyperbolic curves in diagrams involving the incompatible elements and their ratios [68,70]. In the Th vs. Th/Nd and Th/La vs. Zr/Sm diagrams, the studied rocks and associated contemporary mafic rocks, dioritic enclaves, and felsic rocks in the surrounding areas composed a characteristic hyperbolic mixing line (Figure 12), which confirms the major role of a two-component mixing process. To further evaluate the possibility of magma mixing, both Laohushan hornblendite xenoliths (represented by an enriched mantle source) and Quwushan granodiorites (represented by a lower crustal source) were chosen as mixing end-members in the geochemical simulation. The modeling results show that the Caowa high Ba-Sr quartz diorites accord with the formation of crustal and mantle melt mixing, and the mixture proportion was approximately 6:4 (Figure 12b), which was consistent with the Sr-Nd isotope simulation results (Figure 6a).

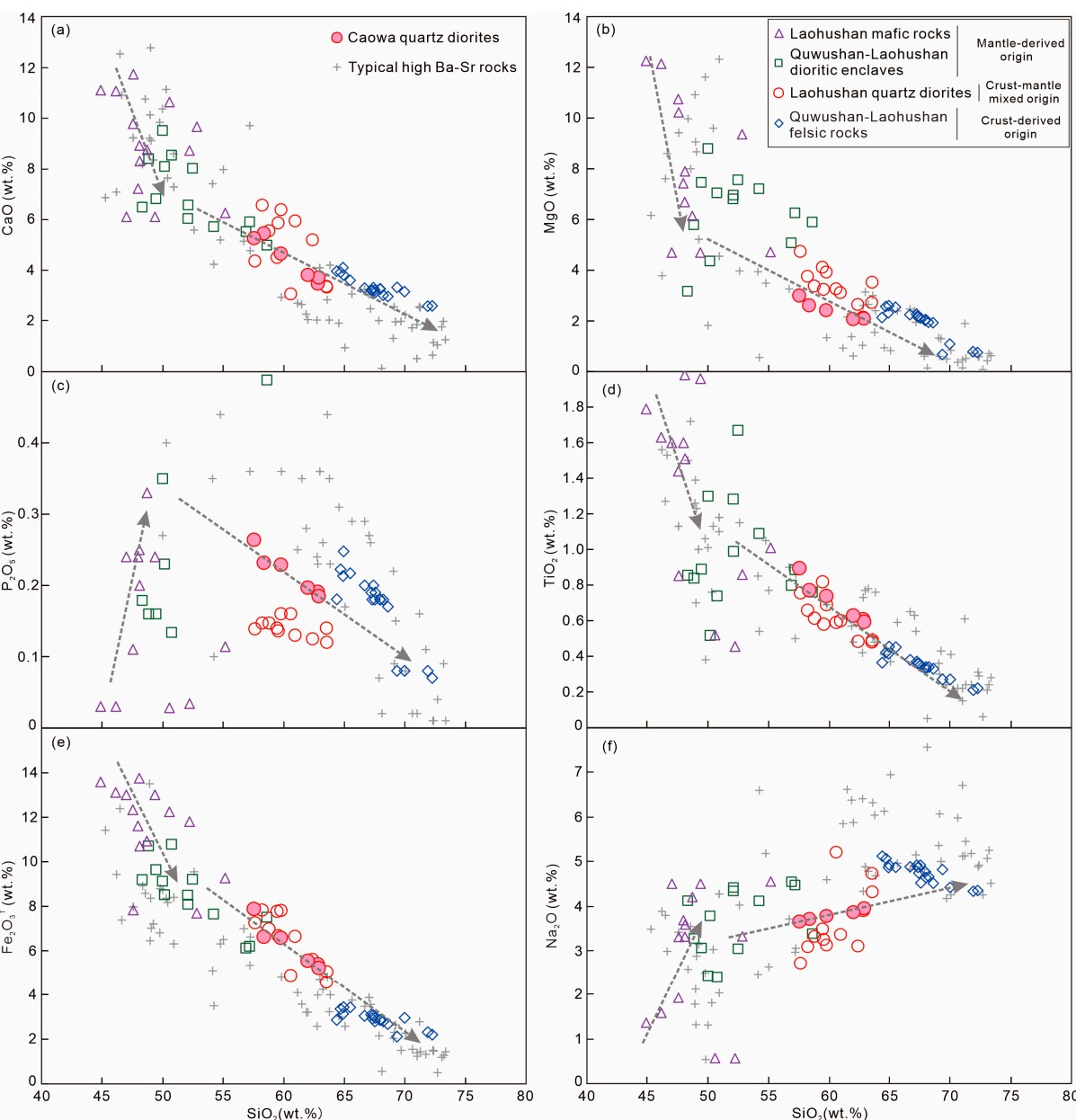

**Figure 11.** Hark diagrams for the Caowa quartz diorites. (**a**) CaO, (**b**) MgO, (**c**) P₂O₅, (**d**) TiO₂, (**e**) Fe₂O₃, (**f**) Na₂O. Data sources: Quwushan-Laohushan mafic rocks, dioritic enclaves, and intermediate-acid rocks are from [13,66,67]. See Figure 4 for other data sources.

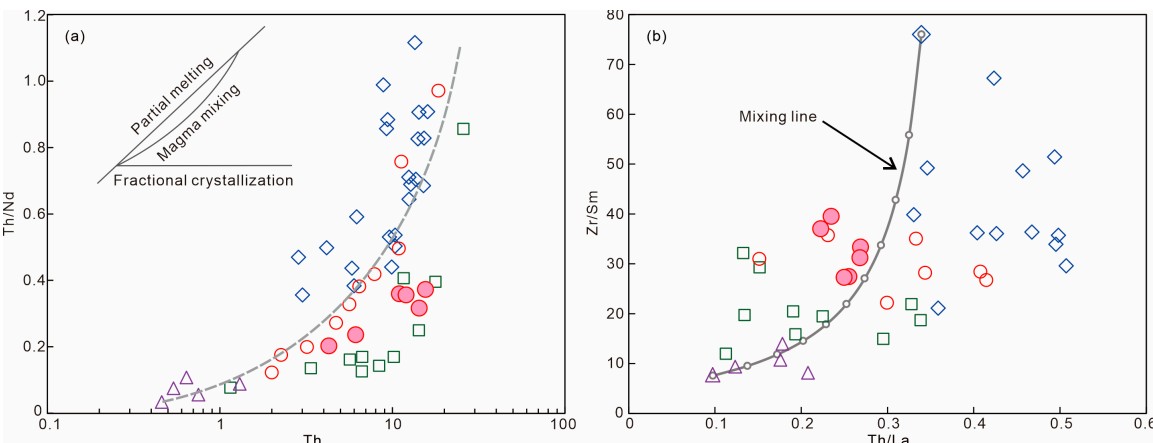

**Figure 12.** Th/Nd vs. Th (**a**) and Zr/Sm vs. Th/La (**b**) diagrams for the Caowa quartz diorites, modified after [70]. See Figure 11 for the symbols and data sources.

In summary, the petrogenesis of Caowa high Ba-Sr intrusion can be explained by a two-stage model: firstly, a low-degree partial melting of enriched lithospheric mantle metasomatized by subduction-related fluids generated initial magmas with high Ba-Sr signatures, and then underplating of this high Ba-Sr basaltic melt triggered partial melting of the ancient lower continental crust and subsequent crust–mantle interaction.

*5.2. Tectonic Implications*

It is generally accepted that the Early Paleozoic NQOB is a typical subduction–accretionary orogenic belt, marking the tectonic evolution of the Proto-Tethys Ocean [1,2,27]. Although both Proto-Tethys Ocean subduction-related and syn-collision/post-collision tectonic settings have been proposed, the subduction polarity and final closure of the Proto-Tethys Ocean are still controversial. The models of the subduction polarity issue include southward subduction [71], northward subduction [5,8], and bidirectional subduction [31,72]. However, considering the current tectonic geographical pattern and the distribution of Early Paleozoic MORB-type ophiolites, arc volcanic rocks, and SSZ-type ophiolites in the NQOB from south to north, northward subduction of the Qilian Proto-Tethys Ocean was basically authenticated, and the closure time of the Qilian Proto-Tethys Ocean was earlier than 440 Ma [5,7,8]. It should be noted that these previous studies concerning the Proto-Tethys evolution have focused on the western part of the NQOB. As for the eastern part of the NQOB, the former evolution process may be "acclimatized" [13], and the closure time of the Qilian Proto-Tethys Ocean maybe earlier than the western part of the NQOB.

Yu et al. (2015) considered that the generation of crustal-derived low-Mg adakitic granitoids (461–440 Ma) from the Eastern NQOB reflects crustal thickening in response to a continent–continent collision [10]. As mentioned above, the Caowa high Ba-Sr dioritic intrusion examined in the Nanhuashan area of the Eastern NQOB was not generated in the partial melting of crustal thickening, which should display high Gd/Yb ratios and leave a residue with garnet. In addition, the distributions of the 448 ± 5 Ma Laohushan ophiolite (the Northern ophiolite belt, Figure 1b) and 446 ± 3 Ma Baiyin arc volcanic rocks indicate that the Eastern NQOB probably underwent a new oceanic crust development and subduction during the Late Ordovician [5,8,66]. Combined with the new discovery of boninitic blueschists and associated greenschists from the Laohushan area in Eastern NQOB by Fu et al. (2022) [26], which records the intra-oceanic subduction initiation at 492–488 Ma, we assume that there may be no continent–continent event occurring there before the Late Ordovician.

The Late Ordovician Caowa high-Ba-Sr quartz diorites studied here are located in the eastern part of the NQOB (Figure 1). As discussed above, these rocks generated through interaction of partial melts derived from subduction-related metasomatized lithospheric

mantle and ancient lower continental crust. Therefore, we propose that they are probably formed in a Late Ordovician oceanic crust subduction-related arc setting. The distribution of these rocks occurring to the northeast of the Baiyin arc and the Laohushan ophiolite belt and the southern margin of Alxa block (Figure 1b) indicates that they were related to the northward subduction of the North Qilian oceanic crust formed in a back-arc basin rather than the paleo-Qilian ocean (maybe a main ocean of Proto-Tethys in the Qilian orogenic system). Consequently, we suggest that, affected by the northward subduction of the Qilian Proto-Tethys Ocean, the Laohushan oceanic crust of the North Qilian back-arc basin was subducted during the Late Ordovician, which induced extensive metasomatism of lithospheric mantle by subduction-related fluids and subsequent crust–mantle interaction during the Late Ordovician (Figure 13).

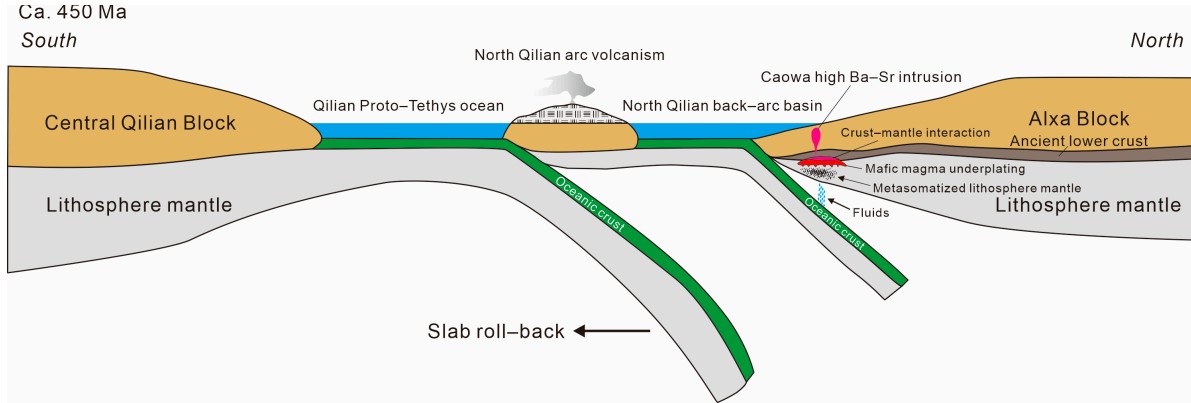

**Figure 13.** Schematic diagram of the possible tectonic setting and petrogenesis of the Caowa high Ba-Sr dioritic intrusion, and the Late Ordovician tectonic evolution of the NQOB.

## 6. Conclusions

(1) Zircon U-Pb dating suggests that the Caowa dioritic intrusion from Nanhuashan area has a magma crystallization age of ca. 450 Ma, representing Late Ordovician magmatism in the eastern North Qilian orogen.

(2) The petrographic, geochemical and Sr-Nd-Hf isotopic characteristics indicate that the Caowa quartz diorites, classified as high Ba-Sr granitoids, were produced through a crust–mantle interaction between partial melt derived from subduction-related fluids metasomatized lithospheric mantle and ancient lower crust-derived magma.

(3) The petrogenesis of the Caowa high Ba-Sr dioritic intrusion in Nanhuashan area provides support for an existence of northward subduction of the North Qilian oceanic crust (Laohushan back-arc oceanic basin) in the eastern North Qilian orogen during the Late Ordovician.

**Author Contributions:** Conceptualization, S.Z. and L.H.; methodology, B.L.; software, H.D. and Q.X.; formal analysis, S.Z.; investigation, S.Z., L.H., and C.M. (Chao Mei); resources, X.W. and C.M. (Caixia Mu); writing–original draft preparation, S.Z.; writing–review and editing, L.H. and B.L.; project administration, L.H. and X.W.; funding acquisition, B.L., H.D., and C.M. (Caixia Mu). All authors have read and agreed to the published version of the manuscript.

**Funding:** This study was co-supported by the National Natural Science Foundation of China (Grant No. 42130309, 41802085), the Provincial Key Research & Development Program of Ningxia Hui Autonomous Region (Grant No. 2021BEG03003), and the Provincial Natural Science Foundation of Ningxia Hui Autonomous Region (Grant No. 2021AAC03447).

**Data Availability Statement:** The original contributions presented in the study are included in the article.

**Acknowledgments:** We deeply appreciate three anonymous reviewers and editor for remarkable constructive comments which significantly improved the manuscript. Jinwei Guo and Cheng Shang are also thanked for assistance of scientific discussions and suggestions in preparing the manuscript.

**Conflicts of Interest:** The authors declare no conflict of interest.

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
