# Peer review of "Late Ordovician High Ba-Sr Intrusion in the Eastern North Qilian Orogen: Implications for Crust–Mantle Interaction and Proto-Tethys Ocean Evolution"

_minerals, doi:10.3390/min13060744_

Round 1
Reviewer 1 Report
1. The main merit of this work lies in its significant contribution to the statistics on geochronology, petrography, geochemistry, and isotopy of rocks in the eastern part of the North Qilian orogenic belt. The accumulation of this particular material will allow (perhaps even other authors) to come to a correct understanding of the geological structure of both this region itself and a larger territory.
2.-3. A geologist studying the area of distribution of rocks of one complex often encounters the fact that some kind of exotic appears among this area. Therefore, the more geological objects of the range are studied, the more reliable the conclusions on the evolution of this region will be.
4. Unfortunately, methodology in geology is a concept limited by various circumstances.
5. Yes, the findings are consistent with the evidence presented. And the presence of previously published works with a similar interpretation of the geology of the region does not detract from the merits of the authors.
6. Links that are in an accessible form are relevant. But there are a number of articles that it is not possible to evaluate.
7. Tables are executed according to the standard scheme. The figures need to be improved (not all signatures are read clearly).
Starting from figure 6, the text is fuzzy. Need to be corrected.

Reviewer 2 Report
The manuscript by Zhao Shaoqing et al. reported high Ba-Sr granitoids, and discuss their petrogenesis and relevant tectonic evolution of Qilian Ocean. The authors suggest that the North Qilian back-arc basin was subducted during the Late Ordovician and resulted in extensive metasomatism of lithospheric mantle by fluids derived from oceanic crust or sediments, and that the Caowa high Ba-Sr quartz diorites were thus generated in the process of crust-mantle interaction during the Late Ordovician. This paper is good quality, and I recommend publication after minor revision.
Minor points:
1. Line 25, delete was;
2. Line 48, delete have been;
3. Fig 1a, change South China Block to South China Plate; North China Craton to North China Plate;
4. Line 87, should be “Central Qilian Block”;
5. Line 202, alternation ? should be alteration;
6. The author are supposed to further explain why the Caowa diorite displays variable εHf(t) values (-13.2 to +8.5). Such high εHf(t) values of +8.5 likely indicate an origin of asthenosphere mantle source, and how to explain the low values of -13.2 and Hf heterogeneities?
The English language looks good but some minor errors.
Reviewer 3 Report
the manuscript presents an extensive quantity of geochemical data with an inset of zircon U-Pb data.
Data are quite well presented, the discussions are nicely linked with the data shown herein and previous works in the area.
the only major flaw concerns the geochronological data, their interpretation, the use of 1s as age error, the graphical plot and the lack of any secondary std (shown in the manuscript) to back up the quality/accuracy of the U-Pb dating.
minor edits in the attached pdf
once this is fixed I reckon the manuscript is good for publication.
all the best

quite easy to read, it flows nicely.
Only few flaws around the manuscript, mostly about lexicon and tenses.
